# Photoperiodic Effect on Growth, Photosynthesis, Mineral Elements, and Metabolome of Tomato Seedlings in a Plant Factory

**DOI:** 10.3390/plants13223119

**Published:** 2024-11-05

**Authors:** Shaofang Wu, Rongguang Li, Chongxing Bu, Cuifang Zhu, Chen Miao, Yongxue Zhang, Jiawei Cui, Yuping Jiang, Xiaotao Ding

**Affiliations:** 1Shanghai Key Laboratory of Protected Horticultural Technology, Horticultural Research Institute, Shanghai Academy of Agricultural Sciences, Shanghai 201403, China; sfwu@saas.sh.cn (S.W.); lrg19971118@163.com (R.L.); zhucuifang1996@163.com (C.Z.); miaochen@saas.sh.cn (C.M.); xuezylemon@foxmail.com (Y.Z.); cuijiawei@saas.sh.cn (J.C.); 2College of Ecological Technology and Engineering, Shanghai Institute of Technology, Shanghai 201418, China; 3Xinjiang Kechuang Tianda Agricultural Engineering Co., Ltd., Changji 831100, China; cxbu66@126.com

**Keywords:** photoperiod, tomato, growth, photosynthesis, mineral elements, metabolome

## Abstract

The duration of light exposure is a crucial environmental factor that regulates various physiological processes in plants, with optimal timing differing between species and varieties. To assess the effect of photoperiods on the growth and metabolites of a specific truss tomato cultivar, three photoperiods (12 h, 16 h, and 20 h) were tested in a plant factory. Growth parameters, including plant height, stem diameter, fresh and dry weights of shoots and roots, photosynthetic characteristics, mineral content, and metabolome profiles, were analyzed under these conditions. The results indicated that prolonged light exposure enhanced plant growth, with the highest photosynthesis and chlorophyll content observed under a 20 h photoperiod. However, no significant correlation was observed between the photoperiod and the mineral element content, particularly for macro minerals. Metabolome analysis revealed that different photoperiods influenced the accumulation of metabolites, particularly in the lipid metabolism, amino acid metabolism, and membrane transport pathways. Long periods of light would enhance photosynthesis and metabolism, improving the rapid growth of tomato seedlings. Overall, this study provides a theoretical basis for understanding the responses of truss tomato cultivars to varying photoperiods in plant factories and proposes an optimizable method for accelerating the progress of tomato seedling cultivation.

## 1. Introduction

Light is a crucial energy source that enables plants to produce food through photosynthesis, thereby sustaining their life processes [1]. Plants can evaluate various light characteristics, such as spectral composition, intensity, duration, and direction, as well as changes in photoperiod, which allow them to regulate their development under different light conditions [2]. Among these factors, photoperiod, referring to the duration of the light period within a 24 h cycle, is a key environmental signal [3]. Plants align their physiological processes with the length of the light period to maximize their growth and reproduction [4]. Adequate photoperiodic sunlight is essential for photosynthesis [5], whereas natural light is often insufficient for optimal plant growth, owing to factors such as clouds, rain, and other climatic conditions. For over a century, supplemental lighting has been used in horticulture to address this limitation [6]. Numerous studies have reported that extending the photoperiod can enhance physiological processes, increase biomass production, and synthesize secondary compounds in plants [7,8,9,10].

The most direct effect of light on plants occurs via photosynthesis. Photosynthesis is a green engine that powers life on Earth because it is the only biological process that allows plants to convert light energy into chemical energy [11]. Prolonging the photoperiod increases light energy input and extends the time available for photosynthesis, leading to greater starch production and biomass yield, as observed in duckweed (*Landoltia punctata*) [12]. Similar results were observed in lettuce and mizuna, where extended photoperiods enhanced growth by increasing light interception and the quantum yield of photosystem II [13]. Additionally, lengthening the photoperiod from 12 to 21 h in a greenhouse increased the chlorophyll content, shoot and root dry mass, plant height, and leaf area of rudbeckia seedlings [14]. Although the maximum photosynthetic rates of lettuce did not increase beyond a 20 h photoperiod, the daily integrated CO_2_ assimilation on a leaf area basis did improve [15].

In addition to influencing photosynthesis and plant growth, photoperiods significantly affect metabolite accumulation [16]. For instance, beet microgreens grown under a 16 h photoperiod demonstrated higher levels of phenolic compounds, total betalains, and antioxidant capacity than those grown under a 12 h photoperiod [17]. Similarly, in red pak choi, tatsoi, and mustard, total phenols and flavonols increased under 20 h and 24 h photoperiods compared with shorter photoperiods [18]. In *Stevia rebaudiana*, the accumulation of antioxidant metabolites increases with light exposure beyond 12 h [19], and longer photoperiods also increase sucrose and starch content in cucumber leaves [20]. However, excessive light, particularly a 24 h photoperiod, can adversely affect plant growth and productivity, causing harm [21]. Continuous lighting has been linked to the development of potentially lethal chlorotic and necrotic spots in the leaves of diverse plant species [22], likely due to an overload of light energy that disrupts the photosynthetic function of chloroplasts [23]. Therefore, determining the optimal photoperiod is essential to maximize crop growth and economic benefits.

Different plants have varying light exposure duration requirements [24], and the effect of photoperiod depends on the cultivar. For example, the celery cultivar ‘Zhangqiubaoqin’ exhibits the highest apigenin content under a 12 h photoperiod, while the ‘Hongchenghongqin’ cultivar reached its peak under a 16 h photoperiod [25]. However, natural light often cannot simultaneously meet the needs of different crops. With the emergence of plant factories, photoperiodic regulation has become feasible. Plant factories, liberated from geographical constraints, maintain precise control over light, temperature, humidity, and carbon dioxide levels to optimize plant growth. This precision agriculture enhances crop yield and quality, as well as enables a consistent annual production [26]. Plant factory can be categorized into sunlight-type and fully artificial light-type factories based on the lighting sources [27]. In the plant factories with artificial lighting (PFALs), light-emitting diodes (LEDs) lamps are widely used as light sources for plant cultivation due to their energy efficiency and precise light supplementation [28,29]. LEDs enable control over light quality, offering a comprehensive spectrum that includes visible light, ultraviolet radiation, and far-red wavelengths. Also, considering the narrow wavelength band emission of LEDs, it is feasible to finely tune the spectral photon flux density profiles to meet the specific needs of various plants.

Tomatoes (*Solanum lycopersicum*) are one of the most important fruit and vegetable crops globally and play a key role in nutritional diets. They are rich in bioactive compounds such as phenolics, flavonoids, carotenoids, vitamins, minerals, and glycoalkaloids, which have attracted significant interest [30]. Tomatoes have a high demand for light, and the duration of light exposure greatly influences their yield and quality. Therefore, it is crucial to investigate their responses to different photoperiods and determine the optimal light environment for various varieties. Although several studies have explored the effects of photoperiod on tomato growth, fruit yield, and development, the results have been inconsistent, likely due to differences in the cultivars studied [31,32,33]. In this study, we focused on truss tomatoes, a popular variety harvested and marketed as bunches with uniform fruit. Using the truss tomato cultivar ‘Glorioso RZ’, we examined the effects of three different photoperiods (12 h/12 h, 16 h/8 h, and 20 h/4 h, light/dark) on plant growth, photosynthesis, mineral content, and metabolome. The goal was to identify the optimal photoperiod for truss tomatoes and provide valuable insights for producing high-quality tomatoes in plant factories with artificial lighting systems.

## 2. Results

### 2.1. Effect of Different Photoperiods on Plant Growth

To analyze the impact of different lighting durations on plant growth, we measured several growth parameters of tomato seedlings under three photoperiods (12 h, 16 h, and 20 h), including plant height, stem diameter, and fresh weight (FW) and dry weight (DW) of the shoots and roots. As shown in Figure 1, both plant height and stem diameter were positively influenced by longer photoperiods, particularly the 20 h photoperiod. Significant differences in plant height and stem diameter were observed. The growth was the highest under the 20 h photoperiod, followed by the 16 h photoperiod, while the plants under 12 h of light demonstrated the poorest growth.

The FW and DW of the tomato shoots and roots were measured at the end of the different photoperiods. Consistent with the trends in plant height and stem diameter, the FW and DW of the shoots and roots increased incrementally with increasing light exposure (Figure 2 and Figure 3). Specifically, compared to the 12 h photoperiod, the FW and DW of tomato shoots under the 20 h photoperiod increased by 213.4% and 235.2%, respectively, whereas the root FW and DW increased by 207.6% and 275.2%, respectively. The 16 h photoperiod resulted in an 82.1% and 82.0% increase in shoot FW and DW, respectively, and a 90.6% and 114.2% increase in root FW and DW, respectively, compared with the 12 h photoperiod (Figure 3). The extension of the photoperiod raised the daily light integral (DLI) levels, which may be directly responsible for the increase in plant biomass. Thus, light energy efficiency was then analyzed according to the plant biomass, which showed that prolonging the photoperiod also improved the utilization efficiency of light energy (Appendix A). Overall, these results indicated that an extended light duration (between 12 and 20 h) promoted significant growth in tomato seedlings.

### 2.2. Effect of Different Photoperiods on Leaf Photosynthesis

Photosynthesis is the primary means by which plants convert light into energy, which directly reflects their growth. Among the three photoperiod treatments, the 20 h photoperiod resulted in the highest net photosynthetic rate (P_n_), stomatal conductance (G_s_), and transpiration rate (T_r_) in tomato leaves (Figure 4A,C,D). However, the intercellular CO_2_ concentration (C_i_) was significantly higher under the 20 h photoperiod compared to the 16 h photoperiod, but showed no difference compared to the 12 h treatment (Figure 4B). Interestingly, the water use efficiency (WUE) was the highest under the 16 h photoperiod, significantly exceeding that of the 12 h and 20 h treatments (Figure 4E).

The duration of light exposure was positively correlated with the relative chlorophyll content in tomato leaves. The highest chlorophyll content was observed under the 20 h photoperiod, followed by the 16 h photoperiod, with the lowest content recorded under the 12 h light exposure (Figure 5).

### 2.3. Effect of Different Photoperiods on Mineral Element Contents

To determine whether the photoperiod affects mineral element content, we analyzed both macro and trace minerals in tomato seedlings under three different photoperiods. The photoperiod treatments had no significant effect on the levels of total nitrogen (TN), phosphorus (P), potassium (K), or magnesium (Mg) (Figure 6A–C,E). However, calcium (Ca) content was reduced under the 20 h photoperiod compared to the 12 h and 16 h treatments (Figure 6D). In contrast, the sulfur (S) content was significantly higher under the 20 h photoperiod compared to the 16 h photoperiod, whereas no significant differences were observed between the 12 h and either the 16 h or 20 h treatments (Figure 6F).

Among the six trace mineral elements analyzed, the contents of iron (Fe) and copper (Cu) in tomato plants were similar across all three photoperiods (Figure 7A,E). The manganese (Mn) content was higher under the 12 h photoperiod than under the 20 h photoperiod, with the 16 h treatment presenting intermediate levels but no significant difference from the other two (Figure 7B). The zinc (Zn) content was higher under the 12 h photoperiod than under the 16 h and 20 h treatments (Figure 7C). Conversely, the boron (B) content was the highest under the 20 h photoperiod, followed by the 16 h treatment, which was higher than the 12 h treatment (Figure 7D). Moreover, the plants under the 16 h photoperiod had more molybdenum (Mo) than those under the 12 h or 20 h treatments, although there was no significant difference between the 12 h and 20 h photoperiods (Figure 7F).

### 2.4. Effect of Different Photoperiods on Metabolome Profiles in Tomato Leaves

To further examine the changes in tomato metabolites under different photoperiods, metabolomic analysis was conducted. The principal component analysis (PCA) score plots showed clear separation among the three samples, with high consistency among replicates in each group (Figure 8A). A heatmap confirmed significant differences in the metabolic spectra of tomato plants across photoperiods, aligned with the PCA results (Figure 8B). Overall, 1587 photoperiod-regulated metabolites exhibited differential abundance changes. Specifically, 949, 1221, and 499 differentially accumulated metabolites (DAMs) were identified in the 16 h vs. 12 h (766 upregulated and 183 downregulated) (Appendix A), 20 h vs. 12 h (803 upregulated and 418 downregulated) (Appendix A), and 20 h vs. 16 h (218 upregulated and 281 downregulated) comparisons (Appendix A), respectively (Figure 8C). Venn diagram analysis identified 141 DAMs common to all three comparisons (Figure 8D and Appendix A). These common DAMs primarily included lipids, lipid-like molecules, organic acids and derivatives, organic oxygen compounds, organoheterocyclic compounds, phenylpropanoids, and polyketides.

To analyze the enrichment of DAMs across the three comparisons, we conducted a Kyoto Encyclopedia of Genes and Genomes (KEGG) analysis. The enrichment was observed in the pathways related to ‘Transport and catabolism’, ‘Membrane transport’, ‘Amino acid metabolism’, ‘Biosynthesis of secondary metabolites’, ‘Carbohydrate metabolism’, ‘Lipid metabolism’, ‘Metabolism of cofactors and vitamins’, ‘Metabolism of other amino acids’, and ‘Nucleotide metabolism’ (Appendix A). The top 20 KEGG terms were selected for further analysis, revealing the differences between comparisons. Nevertheless, five KEGG pathways, including linoleic acid metabolism, glycerophospholipid metabolism, lysine biosynthesis, cysteine and methionine metabolism, and ABC transporters, were enriched in all comparisons (Figure 9). In the linoleic acid metabolism pathway, the levels of PC (18:1/16:1), 7S,8S-DiHODE, and 9S-HpODE decreased with prolonged light exposure (Figure 10A). Similarly, in the glycerophospholipid metabolism pathway, PC (18:1/16:1) and phosphorylcholine levels declined under longer photoperiods (Figure 10B). The amino acids L-homoserine and DL-O-phosphoserine decreased, whereas L-aspartate-semialdehyde increased with extended photoperiods (Figure 10C, D). Furthermore, in the ABC transporter pathway, N, N’-diacetylchitobiose levels were negatively correlated with light duration, while ciprofloxacin levels were positively correlated (Figure 10E). These findings suggest that the photoperiod affects lipid and amino acid metabolism, as well as membrane transport in tomato plants.

## 3. Discussion

### 3.1. Plant Growth and Biomass Are Positively Correlated with Light Duration

Plant growth characteristics can reflect overall plant health, which is closely linked to their ability to resist changing environmental conditions. Great vegetative growth is critical for reproductive growth and stress resistance. As sessile organisms, plants need to coordinate their physiological responses to adapt to the changing environment. Light profoundly affects plant growth and morphogenesis and can fluctuate with time, season, and circadian rhythm. A plant’s response to varying light conditions significantly affects its growth, development, and vitality. Accumulating evidence has suggested that extending the photoperiod promotes plant growth and biomass, as seen by the increased fresh and dry weights of shoots and roots [13,14,20,34,35]. Additionally, plant height, stem diameter, and leaf area tended to increase under longer photoperiods, likely because of enhanced meristem activity and energy utilization [14,15,36]. In this study, we tested three different photoperiods to evaluate the effects of light duration on tomato growth. Consistent with previous research, we discovered that longer photoperiods promoted tomato seedling growth with respect to plant height and stem diameter under the 20 h photoperiod, showing progressively greater differences compared to the shorter photoperiods (Figure 1). Plant biomass, including FW and DW of shoots and roots, was strongly influenced by light duration (Figure 3). However, excessively long photoperiods, or continuous 24 h light, have been shown to suppress plant growth and biomass accumulation and can cause visible physiological disorders such as mottled chlorosis [21,25]. This may be due to an overabundance of light-energy-disrupting chloroplasts as well as the need for darkness for plants to make physiological adjustments and maintain metabolic balance. Numerous studies have suggested that plant growth is closely linked to the DLI, which depends on both light intensity and photoperiod [26,37,38]. In certain crops, such as sugar beet, beetroot, spinach beet, strawberry, lettuce, cabbage, oilseed rape, radish, celery, and tomato, when the same DLI is provided at a lower light intensity over a longer photoperiod, plants can produce more biomass. This is because light drives photosynthesis more efficiently at lower photosynthetic photon flux densities (PPFDs) [39,40,41,42,43]. While this study focused only on the effect of photoperiod on tomato growth, a comprehensive study considering multiple factors, including light intensity, quality, photoperiod, and other environmental conditions, is required to optimize crop cultivation and maximize energy efficiency.

### 3.2. Longer Photoperiod Promotes Photosynthesis of Tomato Seedlings

The relationship between photoperiod and plant growth is primarily driven by photosynthesis, which is the process by which plants capture light energy to assimilate CO_2_ and produce sugars that fuel growth. Photosynthesis involves the two following major stages: light-dependent reactions, which include photosystems and electron transport systems; and light-independent reactions (the Calvin cycle), which are organized into fixation, reduction, and regeneration. This process is critical for plant productivity and is regulated by external environmental signals [44,45]. Several studies have indicated that prolonging the photoperiod stimulates photosynthetic processes [46,47,48], which aligns with our findings. Our results demonstrated that the highest P_n_ occurred under the 20 h photoperiod, which was significantly higher than that under the 12 h and 16 h photoperiods (Figure 4A). Light also regulates stomatal movement in plants [49]. The G_s_, T_r_, and C_i_ of tomato seedlings increased with longer photoperiods from 12 h/16 h to 20 h (Figure 4B–D). However, these parameters were found to decrease in the ‘Micro-tom’ tomato genotype after photoperiod extension [31]. Additionally, transcriptome analysis of pears with varying photoperiods revealed the significant enrichment of photosynthesis-related genes, such as *PbLhca-like* and *PbLhcb-like*, which encode proteins responsible for capturing and transferring light energy [50].

Light harvesting, which is essential for photosynthesis, relies on antenna pigments, primarily chlorophyll, carotenoids, and anthocyanins [51]. Among these, chlorophyll absorbs photosynthetically active radiation most efficiently [52]. Chlorophyll level plays a crucial role in plant photosynthesis. Richardson et al. [53] noted that low chlorophyll concentrations can directly reduce photosynthetic rates, thereby reducing the primary production. The highest leaf chlorophyll content was observed in tomato plants under a 20 h photoperiod, with a positive linear correlation between chlorophyll content and light exposure duration (Figure 5), consistent with the trend in P_n_. Langton et al. [54] hypothesized that this increase in chlorophyll content under longer photoperiods may result from light-dependent chlorophyll synthesis. The increase in chlorophyll with extended photoperiods suggests a combined effect that enables plants to sustain photosynthesis to the light saturation point, promoting greater biomass accumulation. Although the chlorophyll concentration generally increased with extended photoperiods in several species [13,54,55,56,57], some studies have reported that plants grown under short-day conditions contained more chlorophyll per unit leaf area or fresh weight than those cultivated under a 24 h photoperiod. These variations in the effects of photoperiod on plant photosynthesis may be attributed to the differing light requirements and sensitivities of various species.

### 3.3. Mineral Element Contents of Tomato Plants in Response to Different Photoperiods

To sustain photosynthesis, plants require essential mineral elements such as iron (Fe), which plays a crucial role in electron transfer and chlorophyll synthesis, supporting carbon fixation processes [58]. Additionally, mineral elements are important for abiotic stress resistance. For example, K aids in osmotic regulation and enhances salt tolerance [59]. Significant differences in mineral content were observed in vegetables under different photoperiods. Lettuce exposed to a 24 h photoperiod exhibited a significant reduction in the total Zn, Cu, Fe, Mn, Mg, and Ca levels [60]. Similarly, supplementary lighting caused foliar nutrient depletion of N, P, and K in fragrant rosewood, while seedlings subjected to a 3 h extended photoperiod displayed significant symptoms of N and P deficiency [61]. In potatoes, leaves exposed to longer light periods had significantly lower concentrations of all elements, including N, P, K, Ca, Mg, Mn, Fe, Cu, B, and Zn, except Ca, Mg, Mn, and B, which remained consistent across treatments [62]. Similar findings were observed in lettuce, where higher total P and Zn contents were found under shorter photoperiods, whereas total Ca and K contents showed a similar accumulation trend across different photoperiods [63]. Additionally, the contents of most mineral elements, particularly macro minerals, were not significantly affected by varying lighting duration (Figure 6). With the extension of the photoperiod, the B content increased, whereas the Mn content slightly decreased (Figure 7B,D). The extended photoperiod had no significant effect on the overall plant mineral content and, in some cases, even reduced it, likely due to nutrient dilution [64]. As light duration increased, photosynthesis was enhanced, leading to accelerated biomass accumulation, which altered the ratio of elements to dry matter concentration [65].

### 3.4. Photoperiodic Regulation of Metabolome in Tomato

Signaling within plants occurs rapidly in response to varying environmental conditions, resulting in alterations in metabolite synthesis and metabolism [66]. Among multiple environmental factors, the response to light, an essential energy source for plants, is extensive and involves all stages of growth and development [52]. Changes in light conditions, including intensity, quality, and duration, have been widely observed to affect plant metabolite profiles [67,68,69,70]. In this study, we analyzed the metabolic responses of tomato plants to different photoperiods. An extension of the light period to 20 h resulted in significant metabolic changes, totaling 1221 DAMs, compared to the 12 h photoperiod. Additionally, 949 and 499 DAMs were identified in the 16 h vs. 12 h and 20 h vs. 16 h comparisons, respectively (Figure 8C and Appendix A). KEGG enrichment analysis indicated that DAMs in all three comparisons were enriched in lipid and amino acid metabolism (Figure 9 and Appendix A). Furthermore, other studies on tomatoes have demonstrated that changes in the photoperiod can lead to alterations in various metabolites, including phenolic and flavonoid compounds, as well as multiple amino acids [31,71].

Lipids are a class of nonpolar compounds that are relatively insoluble in water [72] and serve as the backbone of membranes, thereby providing essential biomolecular layers for plants. In addition to their structural roles, lipids can function as signaling molecules and biological effectors. It has been established for two decades that light can stimulate fatty acid and lipid formation in leaves up to 25 times [73]. An increase in polyunsaturated fatty acid-containing lipids was observed in broccoli sprouts grown under light/dark cycles compared with those grown in continuous darkness [74]. Lipid accumulation is a photoperiodically controlled trait, as evidenced by the finding that cells of *Chlamydomonas reinhardtii* accumulate more lipids under short-day conditions than under long-day conditions. Silencing the *CrCO* gene, which regulates photoperiod, also increased lipid content [75]. A similar phenomenon was noted in hybrid aspen, which accumulated more lipids in response to short-day exposure [76]. In our study, several metabolites associated with linoleic acid and glycerophospholipid metabolism accumulated to higher levels at a 12 h photoperiod in tomato plants compared to 16 h and 20 h photoperiods (Figure 10A,B), whereas the levels of 13S-HODE and 12R,13S-epoxy-9S-hydroxy-10E-octadecenoic acid peaked at a 16 h photoperiod compared to the 12 h and 20 h photoperiods (Figure 10B). The accumulation of different types of fatty acids in *Chlorella vulgaris* varied under different photoperiods, with the total saturated fatty acids increasing and monounsaturated and polyunsaturated fatty acids decreasing as light duration increased [77]. Additionally, a dominant photoperiod- and thermo-sensitive dwarf mutant, *Photoperiod-thermo-sensitive dwarfism 1 (Ptd1)*, has been identified, which loses the function of the gene encoding a non-specific lipid transfer protein (nsLTP). *Ptd1* plants exhibited severe dwarfism under long-day and low-temperature conditions but grew almost normally under short-day and high-temperature conditions. These findings suggest a close relationship between photoperiod and lipid metabolism.

Amino acids are key components of living organisms and serve as the primary units of proteins for body construction, as well as energy sources, chemical messengers, and precursors for diverse metabolites [78]. Increasing evidence indicates that amino acid biosynthesis and metabolism undergo plastic alterations to coordinate specific growth and developmental events. As a major environmental factor throughout the plant life cycle, the effects of light on amino acid synthesis and metabolism have been gradually explored. In *Trigonella* microgreens, the concentrations of essential amino acids, including histidine, isoleucine, leucine, phenylalanine, threonine, valine, and lysine, and non-essential amino acids, such as aspartic acid, glutamic acid, glycine, arginine, alanine, tyrosine, asparagine, and glutamine, significantly decreased in response to a longer photoperiod. Similarly, the concentration of cysteine, a sulfur-containing amino acid, decreases in response to a prolonged exposure to light [79]. Higher amino acid production was observed under a 16 h photoperiod compared to an 8 h photoperiod in *Piper nigrum* [80]. In a study on *Allium sativum*, transcriptome analysis evaluated the transcriptional response to different photoperiods, revealing DEGs enriched in pathways related to amino acid biosynthesis [81]. In this study, the metabolome analysis of tomato plants under three different photoperiods indicated that DAMs in all three comparisons were enriched in amino acid metabolism, including lysine biosynthesis, and cysteine and methionine metabolism (Figure 9 and Appendix A). Specifically, L-homoserine, L-saccharopine, and DL-O-phosphoserine accumulated more under the 12 h photoperiod than under the two longer photoperiods, whereas the contents of L-aspartate-semialdehyde and 5-methylthioribose were highest under the 20 h photoperiod (Figure 10). This finding is consistent with that of another study on rice, where DAMs were enriched in the amino acid pathway in the long-day vs. short-day comparison, and asparagine, pyridoxamine, and pyridoxine accumulated at higher levels in rice grains harvested from the short-day photoperiod [70]. Further investigation of the specific mechanisms by which photoperiod affects metabolite accumulation is warranted.

## 4. Materials and Methods

### 4.1. Plant Material and Treatments

The experiment was conducted in an artificial light plant factory at the Chongming base of the National Engineering Research Centre of Protected Agriculture (31°34′ N, 121°41′ E), Shanghai Academy of Agriculture Sciences, China between 2022 and 2023. The tomato variety adopted in this study was Glorioso RZ (Rijk Zwaan Company, De Lier, The Netherlands). The tomato seeds were sown in coconut chaff cubes (soaked in a nutrient solution with an EC of 3.0 ds/m and a pH of 5.5 for one day, with the size after soaking approximately 10 cm × 10 cm × 7.5 cm) on 7 December 2022. The 7-day-old tomato seedlings were subjected to different photoperiodic treatments for 20 days.

Three photoperiodic treatments were implemented in the plant factory as follows: (1) 12 h light/12 h dark, (2) 16 h light/8 h dark, and (3) 20 h light/4 h dark, with 16 plants per treatment replicated three times. The temperature during the light/dark period was maintained at approximately 26 ± 2 °C/17 ± 2 °C, with relative humidity ranging from 60% to 90%, the nutrient solution was supplied with an EC of 3.0 ds/m and a pH of 5.5, and no CO_2_ enrichment. An appropriate fresh air exchange was provided during the day to enhance the CO_2_ supply, and a recirculation fan was continuously operated. The PPFD at the canopy apex of the 7-day-old seedlings, measured 15 cm from the LED light source, was approximately 242.72 µmol·m^−2^·s^−1^ (380–780 nm), with 123.07 µmol·m^−2^·s^−1^ for red light (600–700 nm), 63.91 µmol·m^−2^·s^−1^ for blue light (400–499 nm), 51.43 µmol·m^−2^·s^−1^ for green light (500–599 nm), 4.13 µmol·m^−2^·s^−1^ for far-red light (701–780 nm), and 0.18 µmol·m^−2^·s^−1^ for ultraviolet light (380–399 nm) (Table 1). The DLI values of the 12 h, 16 h, and 20 h photoperiod treatments were 10.49 mol·m^−2^·d^−1^, 13.98 mol·m^−2^·d^−1^, and 17.48 mol·m^−2^·d^−1^, respectively (Table 2).

### 4.2. Measurement of Growth Parameters

After the 20-day treatment, plant height and stem diameter of the tomato seedlings under different photoperiods were measured using a ruler and Vernier caliper, respectively. Then, the shoots and roots of the plants were collected and weighed as FW, and then dried at 80 °C for 3 d to measure DW. The utilization efficiency of light energy was calculated according to the DW of the whole plant and the DLI values. Three biological replicates were used for each treatment.

### 4.3. Measurement of Relative Chlorophyll Content

The relative chlorophyll content of the leaves was measured on the 20th day of treatment using a portable chlorophyll meter (SPAD 502, Minolta, Tokyo, Japan). Measurements were conducted on the largest fully developed leaves, and each treatment measurement was replicated at least five times [82].

### 4.4. Measurement of Photosynthesis

On the 20th day of treatment, gas exchange parameters including P_n_, G_s_, C_i_, T_r_, and WUE were measured using a portable photosynthesis system (CIRAS-3; PP Systems, Amesbury, MA, USA). The measurements were conducted on the largest newly expanded upper-developed leaves, with the light intensity set at 1000 µmol·m^−2^·s^−1^, whereas the air temperature, relative humidity, and CO_2_ concentration were dependent on the natural conditions of the plant factory. Five biological replicates were used for each treatment group.

### 4.5. Measurement of Mineral Element Contents

Tomato seedlings were heated at 105 °C for 2 h and subsequently dried at 80 °C for 3 d. The mineral element contents of the tomato plants were analyzed by Eurofins Analytical Services Co. (branch of Suzhou, China), following the modified methods described by Song et al. [83]. Total nitrogen was analyzed using the near-infrared method, whereas Mo was determined using inductively coupled plasma mass spectrometry (ICP-MS). The P, K, Ca, Mg, Zn, S, Fe, Mn, B, and Cu contents were assessed by flow analysis using inductively coupled plasma atomic emission spectrometry (ICP-AES). Three biological replicates were used for each treatment group.

### 4.6. Metabolite Profiling and Data Analysis

Tomato leaves were collected 20 d after the different photoperiod treatments, with each treatment consisting of six biological replicates. A sample of 60 mg was ground with 70% methanol after pre-cooling for 2 min, ultrasonicated in an ice-water bath for 30 min, and stored at −40 °C for 2 h. After centrifugation for 20 min at 13,000 rpm and 4 °C, 150 μL of the supernatant was transferred to LC vials for LC-MS analysis. Metabolomic analysis was conducted using an ultra-performance liquid chromatography-tandem mass spectrometry (UPLC-MS/MS) system (Waters ACQUITY UPLC I-Class plus/Thermo QE plus) equipped with a 1.8 μm column (ACQUITY UPLC HSS T3, 100 mm × 2.1 mm). The column temperature was maintained at 45 °C, and the mobile phase consisted of 0.1% formic acid in water (A) and acetonitrile (B), with a flow rate of 0.35 mL/min and an injection volume of 3 µL. The gradient consisted of 95%, 70%, 50%, 20%, and 0% A for 2, 4, 2, 4, and 1 min, respectively, followed by an increase back to 95% in 0.1 min, with a 1-minute re-equilibration period. The mass spectrometer was operated in both negative and positive ionization modes, with the ESI source conditions set to a mass range of 70–1050 *m*/*z*, an ion source temperature of 320 °C, and an ion spray voltage of 3800 V (positive ion mode) and −3000 V (negative ion mode).

Raw data were analyzed using Progenesis QI v3.0 software (Nonlinear Dynamics, Newcastle, UK) for baseline filtering, peak identification, integration, retention time correction, peak alignment, and normalization. Annotation of the identified metabolites was conducted using the Human Metabolome Database (HMDB), Lipidmaps (v2.3), METLIN database, and LuMet-Plant3.0 local database. PCA and partial least squares discriminant analysis (PLS-DA) were performed using MetaX. DAMs were defined as those with a *p*-value < 0.05 and FC ≥ 2.0 or ≤0.5. Pathway enrichment analysis was performed using the KEGG database.

### 4.7. Statistical Analysis

Statistical analysis of the bioassay data was conducted using SAS version 9.3 software (SAS Institute Inc., Cary, NC, USA). The results were presented as mean ± standard deviation (SD) with a minimum of three replicates. Differences among treatment means were analyzed using the Least Significant Difference (LSD) method at a significance level of *α* = 0.05, with significant differences indicated by different letters in the figures. The figures were plotted using Origin 7.5 (Origin Lab, Northampton, MA, USA).

## 5. Conclusions

Photoperiods exert diverse regulatory effects on numerous physiological processes in plants, ultimately influencing the yield and quality of vegetables. This study determined the effects of three different photoperiods on the growth, photosynthesis, mineral element content, and metabolites of tomato seedlings. Our findings confirmed that a suitable photoperiod extension promoted plant growth, as evidenced by increases in the plant height, stem diameter, and fresh and dry weights of both shoots and roots. Moreover, the leaves of plants under the 20 h photoperiod exhibited the highest photosynthetic rates, with enhanced P_n_, C_i_, G_s_, and T_r_. Consistent with these photosynthetic characteristics, the chlorophyll content was the highest in the plants under the 20 h photoperiod, followed by the 16 h photoperiod, while the 12 h photoperiod resulted in the lowest chlorophyll levels. However, the mineral element content was minimally affected by light duration, particularly for macro elements, such as N, P, and K, which maintained consistent levels across different photoperiods. Furthermore, metabolic analysis indicated that the enriched DAMs under various photoperiods were predominantly identified in lipid metabolism, amino acid metabolism, and membrane transport pathways. These findings could be beneficial for cultivating robust tomato seedlings and provide theoretical guidance and references for production.

## Figures and Tables

**Figure 1 plants-13-03119-f001:**
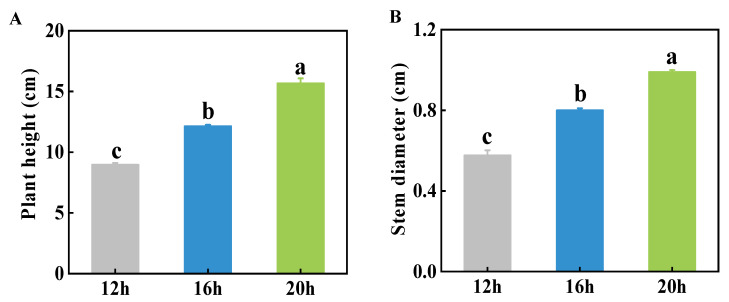
Effect of three different photoperiods (12 h, 16 h, and 20 h) on plant height (**A**) and stem diameter (**B**) of tomato seedlings. The data represent the mean of at least three biological replicates. Different letters indicate significant differences (LSD) (*α* = 0.05).

**Figure 2 plants-13-03119-f002:**
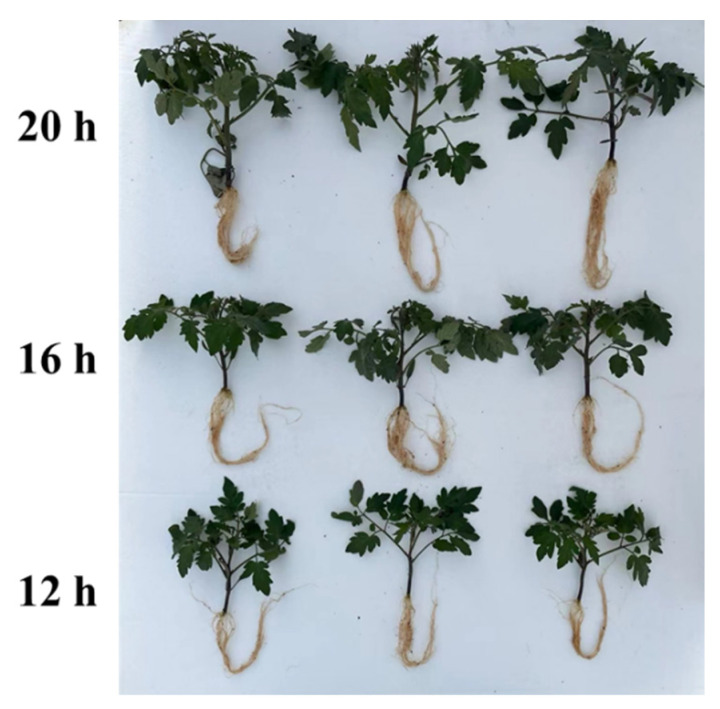
Effect of three different photoperiods (12 h, 16 h, and 20 h) on tomato seedlings. After 20 days of treatment, the plant seedlings were photographed.

**Figure 3 plants-13-03119-f003:**
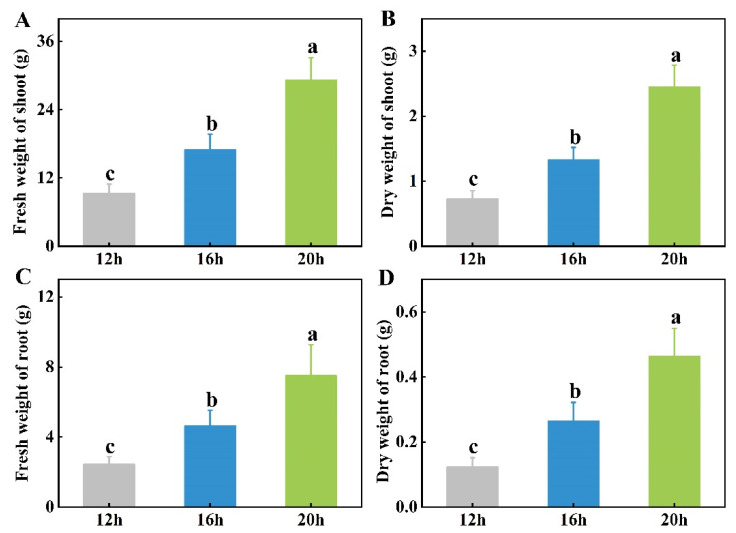
Effect of three different photoperiods (12 h, 16 h, and 20 h) on fresh and dry weights of shoots (**A**,**B**) and roots (**C**,**D**). The data represent the mean of at least three biological replicates. Different letters indicate significant differences (LSD) (*α* = 0.05).

**Figure 4 plants-13-03119-f004:**
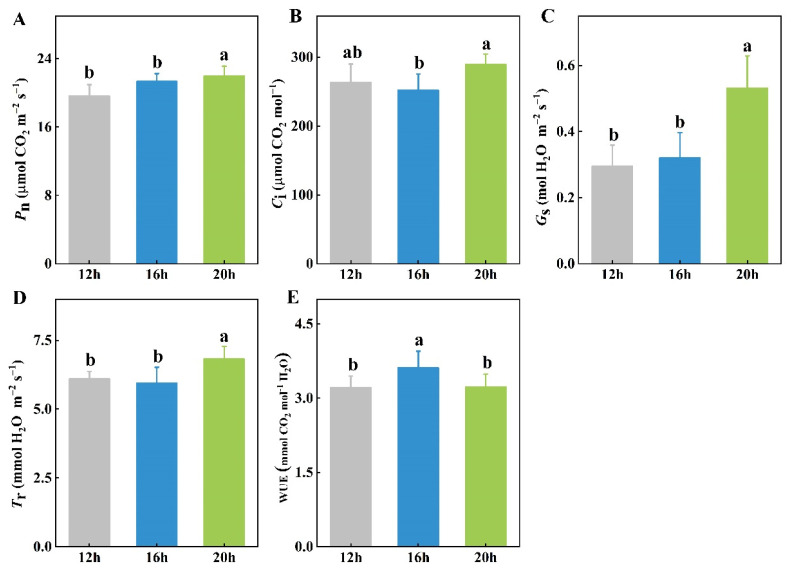
Effects of three different photoperiods (12 h, 16 h, and 20 h) on the net photosynthetic rate (P_n_) (**A**), intercellular CO_2_ concentration (C_i_) (**B**), stomatal conductance (G_s_) (**C**), transpiration rate (T_r_) (**D**), and water use efficiency (WUE) (**E**) of tomato leaves. The data represent the mean of at least three biological replicates. Different letters indicate significant differences (LSD) (*α* = 0.05).

**Figure 5 plants-13-03119-f005:**
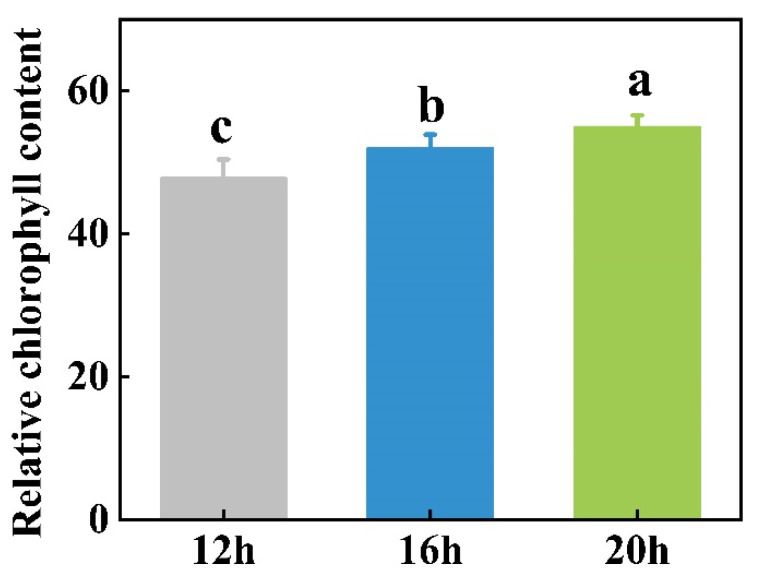
Effects of three different photoperiods (12 h, 16 h, and 20 h) on the relative chlorophyll content of tomato leaves. The data represent the mean of at least three biological replicates. Different letters indicate significant differences (LSD) (*α* = 0.05).

**Figure 6 plants-13-03119-f006:**
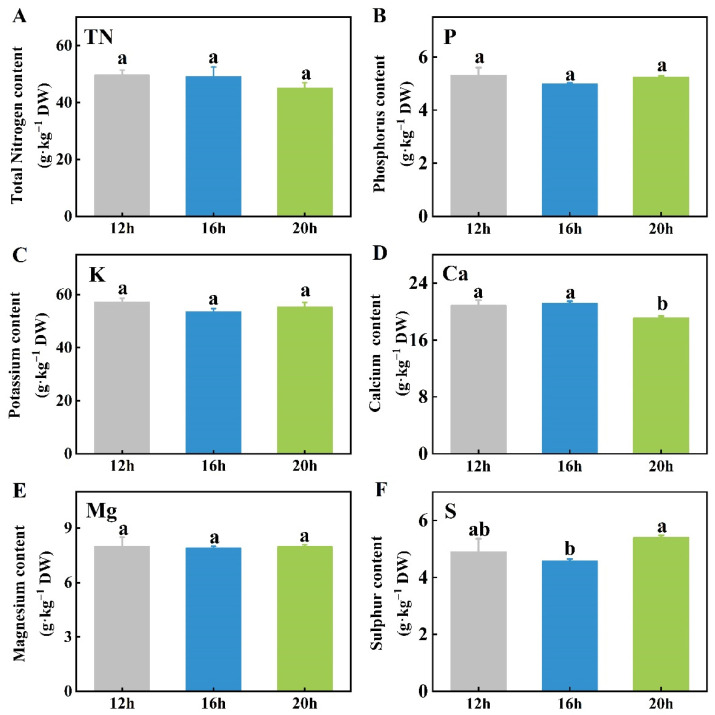
Effects of three different photoperiods (12 h, 16 h, and 20 h) on the contents of six macro mineral elements, TN (**A**), P (**B**), K (**C**), Ca (**D**), Mg (**E**), and S (**F**), in tomato shoots. The data represent the mean of at least three biological replicates. Different letters indicate significant differences (LSD) (*α* = 0.05). TN, total nitrogen; P, phosphorus; K, potassium; Ca, calcium; Mg, magnesium; S, sulfur.

**Figure 7 plants-13-03119-f007:**
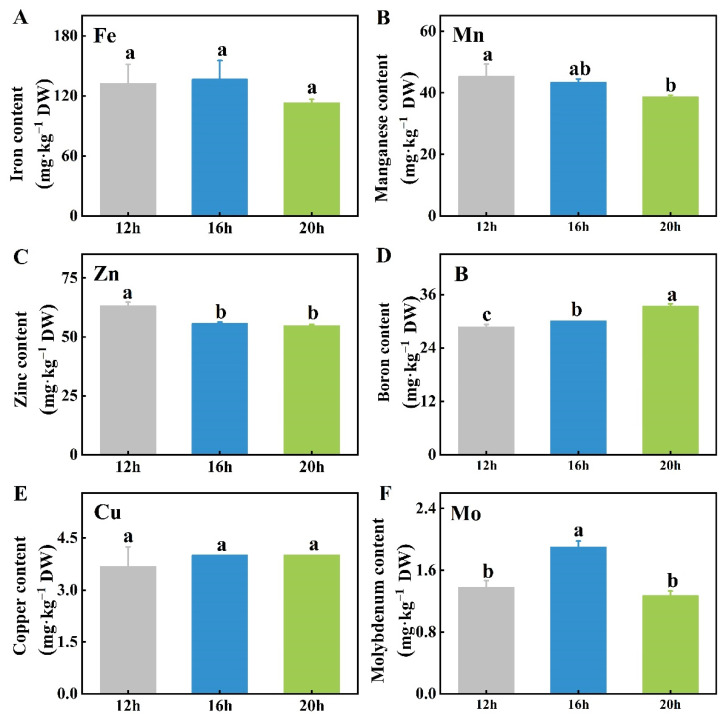
Effects of three different photoperiods (12 h, 16 h, and 20 h) on the contents of six trace mineral elements, Fe (**A**), Mn (**B**), Zn (**C**), B (**D**), Cu (**E**), and Mo (**F**), in tomato shoots. The data represent the mean of at least three biological replicates. Different letters indicate significant differences (LSD) (*α* = 0.05). Fe, iron; Mn, manganese; Zn, zinc; B, boron; Cu, copper; Mo, molybdenum.

**Figure 8 plants-13-03119-f008:**
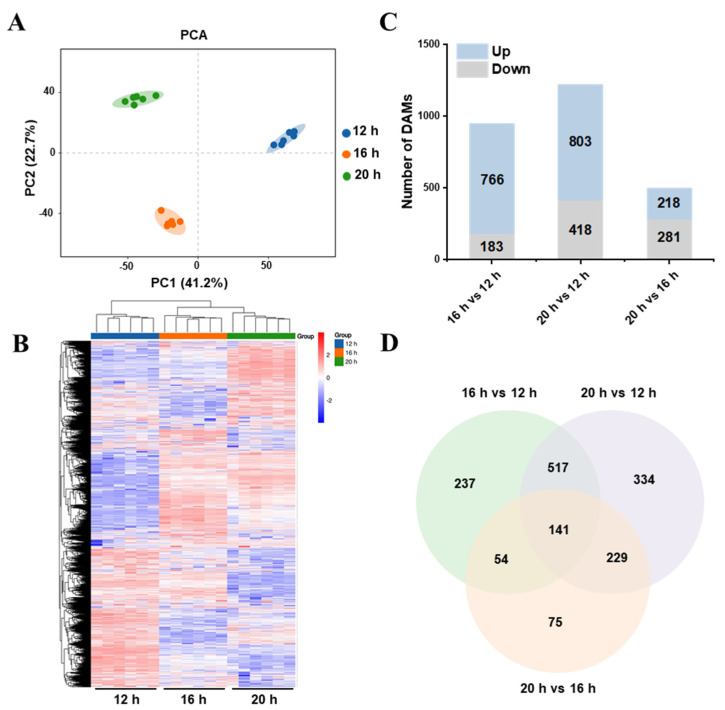
Metabolome analysis of tomato seedlings under different photoperiods. (**A**) Principal component analysis (PCA) of metabolites under three different photoperiods. Circles represent 95% confidence intervals. (**B**) Cluster heat map of relative metabolite content in tomato plants under three different photoperiods. (**C**) The total number of upregulated and downregulated differentially accumulated metabolites (DAMs) in the different groups. (**D**) Venn diagram of the DAMs under three different photoperiods.

**Figure 9 plants-13-03119-f009:**
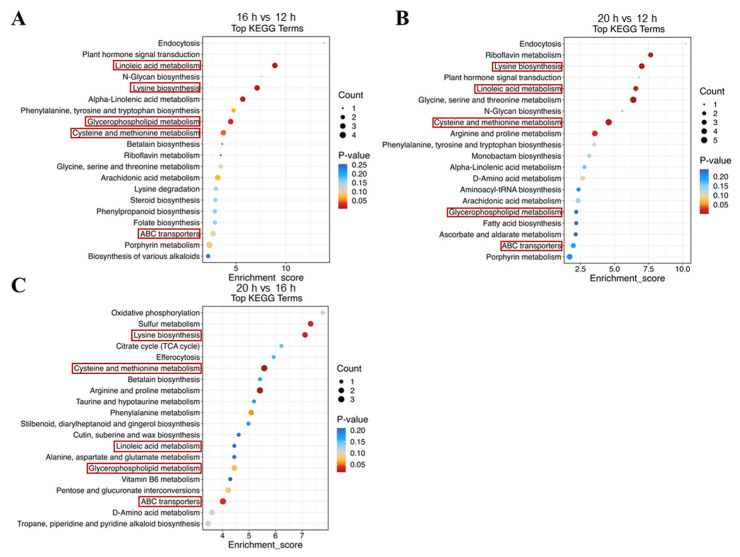
Kyoto Encyclopedia of Genes and Genomes (KEGG) enrichment of differentially accumulated metabolites in the comparison of 16 h vs. 12 h (**A**), 20 h vs. 12 h (**B**), and 20 h vs. 16 h (**C**). The shared KEGG pathways in three comparisons were marked with red frame.

**Figure 10 plants-13-03119-f010:**
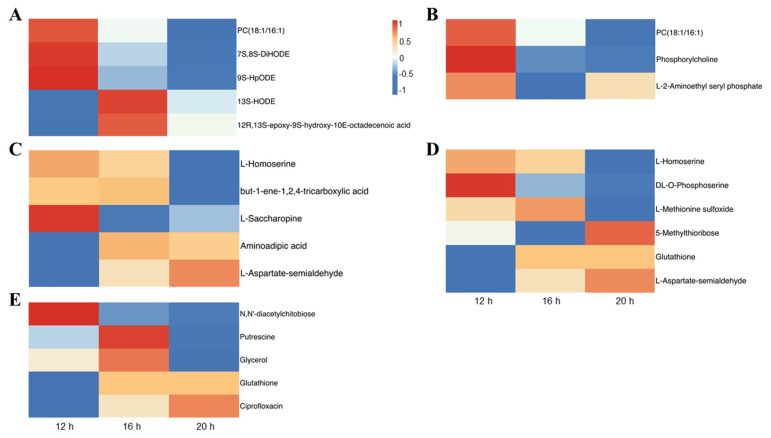
Patterns of accumulation of differentially accumulated metabolites involved in linoleic acid metabolism pathway (**A**), glycerophospholipid metabolism pathway (**B**), lysine biosynthesis pathway (**C**), cysteine and methionine metabolism pathway (**D**), and ABC transporter pathway (**E**).

**Table 1 plants-13-03119-t001:** PPFD radiation parameters of seedlings at the beginning of treatment.

Spectral Ranges (nm)	Total (380–780)	UV (380–399)	Blue (400–499)	Green (500–599)	Red (600–700)	Far-Red (701–780)
PPFD (µmol·m^−2^·s^−1^)	242.72	0.18	63.91	51.43	123.07	4.13

Note: Measured at 15 cm away from the LED tube.

**Table 2 plants-13-03119-t002:** Daily light integral (DLI) of different photoperiod treatments.

Photoperiod	12 h	16 h	20 h
DLI (mol·m^−2^·d^−1^)	10.49	13.98	17.48

## Data Availability

The data presented in this study are available on request from the corresponding author.

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
