# Peer review of "Photoperiodic Effect on Growth, Photosynthesis, Mineral Elements, and Metabolome of Tomato Seedlings in a Plant Factory"

_plants, 2024, doi:10.3390/plants13223119_

Round 1

Reviewer 1 Report

Comments and Suggestions for Authors

Review of the article by Shaofang Wu, Rongguang Li, Chongxing Bu, Cuifang Zhu, hen Miao, Yongxue Zhang, Jiawi Cui, Yuping Jiang and Xiaotao Ding: "Photoperiodic Effect on Growth, Photosynthesis, Mineral Elements, and Metabolome of Tomato Seedlings in a Plant Factory"

 The paper evaluates the effect of different light periods on the growth and metabolism of a particular tomato variety. Three photoperiods were tested (12 h, 16 h and 20 h. The results showed that prolonged exposure to light contributes to increased plant growth. The work allows us to assess the orientation of a number of tomatoes to change the photoperiod and suggests the optimal method to accelerate the process of growing tomato seedlings.

There are a number of questions about the article. Thus, according to Figures 3,4,5, the question arises about the reliability of the results, since Figure 6 shows that the differences between the options are usually not reliable. Rather, it is necessary to talk about the tendency to increase or decrease certain indicators.

In the work, it makes sense to calculate the energy efficiency of using light energy according to the experience options. This will answer whether it makes sense to increase the light period from the point of view of the economic component, since an increase in the photoperiod in the light culture of plants is often accompanied by a decrease in light intensity.   The article can be accepted for publication after some revision.

Reviewer 2 Report

Comments and Suggestions for Authors

The manuscript is prepared correctly, I have no comments.

Reviewer 3 Report

Comments and Suggestions for Authors

plants-3272388

Comments to authors,

The authors investigated the effects of day and night lighting duration on the dry matter content, physiological and metabolic processes of tomato seedling plants. Despite the use of the term plant factory in the title, there is a little mention of its application or type in tomato production in the introduction section. Figures and tables are used to illustrate the effects of different lighting durations on plants, but the methodological chapter needs a detailed description of the experiment, and the presentation of the lighting conditions needs improvement. The reference list is inaccurate, with some missing journal names.

Detailed comments

Complete the literature review section with where, for what purpose and how are "plant factories" used in crop production. Include the type of illumination and the composition of the light spectrum.

In some figures, it is necessary to check the significance of differences between treatments:

-7A in Figure 7, it seems to me that the Fe content is lower in the 20hr treatment than in the 16hr treatment, it should be checked.

Line 181: "The boron content of the shoot was higher in the 16hr treatment than in the 12 hr treatment". There is no significant difference in boron content between the 12 and 16 hr treatments, see Figure 7D. Check the letter notations in Figure 7D and the conclusion (line 181) and the discussion section in lines 324-325.

Line 241: What is robust vegetative growth? The claim is debatable, reproductive growth is influenced by many factors. Rephrase the sentence.

Lines 242-243: This sentence is not clear. Check and correct.

In the methodological chapter, when describing the experiment, not only the photoperiodic treatments but also the lighting conditions and nutrient supply should be described in more detail.

The experimental set-up and design should be extended, e.g. What was the age of the plants when the treatments were begun? How long did the treatment last on the plants? Few plants are shown in Figure 11. How many plants were measured and used for chemical analysis?

Lines 413-418: I don’t understand the sentence exactly. How was the seedlings lighted by continuously or pulsating?  It appears that white LED light sources were supplemented with different LED light sources. It would be better to provide a table with the properties of the treatments, including LED spectrum, PPFD, DLI values, to make it easier to understand.

The seedlings size shown in Figure 11 cannot be valuated and there is no explanation for the lighting which should be explained or the figure deleted.

Line 445: The full name of the TN abbreviation must also be given.

In the reference list, the source is missing for the following:

6,10,21,28,67 references are missing the name of the journal.

Round 2

Reviewer 3 Report

Comments and Suggestions for Authors

Revised  plants 327388

Authors taking into account the suggestions have improved the manuscript. I accept the responses in Figures 7A and D for marking significant differences.

However, an error was made in the reference list during the corrections. References 21 and 22 are identical and should be corrected.